# Diagnostic Performance of Electromagnetic Navigation Bronchoscopy-Guided Biopsy for Lung Nodules in the Era of Molecular Testing

**DOI:** 10.3390/diagnostics11081432

**Published:** 2021-08-09

**Authors:** Ju Hyun Oh, Chang-Min Choi, Seulgi Kim, Se Jin Jang, Sang Young Oh, Mi Young Kim, Hee Sang Hwang, Wonjun Ji

**Affiliations:** 1Asan Medical Center, Department of Pulmonary and Critical Care Medicine, University of Ulsan College of Medicine, Seoul 05505, Korea; angle5018@naver.com (J.H.O.); ccm@amc.seoul.kr (C.-M.C.); seulgi@amc.seoul.kr (S.K.); 2Asan Medical Center, Department of Oncology, University of Ulsan College of Medicine, Seoul 05505, Korea; 3Asan Medical Center, Department of Pathology, University of Ulsan College of Medicine, Seoul 05505, Korea; jangsj2644@gmail.com; 4Asan Medical Center, Department of Radiology, University of Ulsan College of Medicine, Seoul 05505, Korea; ojangsa@gmail.com (S.Y.O.); mimowdr@gmail.com (M.Y.K.)

**Keywords:** accuracy, diagnostic yield, electromagnetic navigation bronchoscopy, lung nodule, safety

## Abstract

Electromagnetic navigation bronchoscopy (ENB) is an emerging technique used to evaluate peripheral lung lesions. The aim of this study was to determine the diagnostic yield, safety profile, and adequacy of specimens obtained using ENB for molecular testing. This single-center, prospective pilot study recruited patients with peripheral pulmonary nodules that were not suitable for biopsy via percutaneous transthoracic needle biopsy methods. The possibility of molecular testing, including epidermal growth factor receptor (EGFR), anaplastic lymphoma kinase (ALK), and programmed death ligand 1 (PD-L1), was identified with non-small cell lung cancer (NSCLC) tissue obtained using ENB. ENB-guided biopsy was performed on 30 pulmonary nodules in 30 patients. ENB-guided biopsy was successfully performed in 96.6% (29/30) of cases, but one case failed to approach the target lesion. The diagnostic accuracy of ENB-guided biopsy was 68.0% (17/25). Biopsy-related pneumothorax occurred in one patient and there was no major bleeding or deaths related to the procedure. Among 13 patients diagnosed with NSCLC, molecular testing was successfully performed in 92.3% (12/13). ENB-guided biopsy demonstrated acceptable accuracy and excellent sample adequacy, with a high possibility of achieving molecular testing and a good safety profile to evaluate peripheral pulmonary nodules, even when the percutaneous approach was difficult and/or dangerous.

## 1. Introduction

The recent emergence of the need for lung cancer screening has prompted the increased use of chest computed tomography (CT), which in turn has increased the detection of lung nodules [1,2]. Because lung nodules suspicious for lung cancer are frequently found in heavy smokers with underlying lung diseases, such as emphysema, chronic obstructive pulmonary disease [3], or interstitial lung disease [4], more accurate and less invasive modalities are required for biopsy. Moreover, finding target molecules, such as epidermal growth factor receptor (EGFR), anaplastic lymphoma kinase (ALK), ROS1, and programmed death ligand 1 (PD-L1), is significantly important for making treatment decisions in patients with lung cancer. As such, it is necessary to perform molecular testing at the time of initial diagnosis [5].

Electromagnetic navigation bronchoscopy (ENB) is an image-based, real-time navigation system that creates virtual three-dimensional images of the tracheobronchial tree, and it enables operators to approach peripheral lung nodules with a bronchoscopy catheter that could not be accessed using percutaneous methods or surgical biopsy due to a high risk of procedure-related complications [6]. The diagnostic yield of ENB has been reported to be variable, ranging from 60% to 94% according to the type of sedation and the combined diagnostic tools including radial endobronchial ultrasound (r-EBUS), positron emission tomography-CT (PET-CT), and rapid on-site examination (ROSE), among others [6,7,8,9,10]. Despite many studies having reported the diagnostic performance and safety of ENB in Western countries, which developed the ENB technique [11,12,13], the prospective study of ENB in Asia where most endoscopies are conducted under moderate sedation is not sufficient. Notably, in South Korea, since ENB equipment has only recently been introduced, clinical research data on the accuracy and safety of ENB-guided biopsy for peripheral lung lesions are scarce. Moreover, when considering that most of the existing ENB reports are retrospective studies based on the first generation ENB model, the superDimension Navigation System (Medtronic, Minneapolis, Minnesota), studies based on the second generation ENB model, the SPiN Thoracic Navigation System(Veran Medical, St. Louis, MO, USA), are also needed.

In this prospective study, we aimed to identify the diagnostic yield and safety of ENB-guided biopsy using the second generation ENB model, the SPiN Thoracic Navigation System, for patients with high-risk peripheral lung nodules in whom it was not possible to obtain tissue using percutaneous transthoracic needle biopsy (PCNB), under moderate sedation. Furthermore, we investigated whether the small biopsy specimen obtained using ENB was sufficient for the proper molecular testing that is important in treating lung cancer.

## 2. Materials and Methods

### 2.1. Subjects and Data Collection

The present investigation was a prospective, single-center study. Thirty consecutive patients, who underwent ENB between November 2017 and October 2018 at Asan Medical Center (Seoul, Republic of Korea), were enrolled. The inclusion criteria for ENB were patients who exhibited suspicious peripheral lung nodules on chest CT, but could not undergo PCNB or other invasive methods due to high risks of complications such as bleeding and pneumothorax, or an inaccessible location for biopsy. The peripheral lung nodules on chest CT were reviewed for the possibility of PCNB for target lesions by radiologists and pulmonologists before performing ENB. Patients who were over 80 or under 18 years old, and those who refused ENB, had a recent history of acute coronary syndrome, stroke, or uncontrolled arrhythmia, or had an endobronchial lesion on bronchoscopy, were excluded from the study. The study protocol was approved by the Institutional Review Board of Asan Medical Center (approval no. 2017-0196), and informed consent was obtained from all subjects. Demographic data, including age, sex, smoking habits, the results of pulmonary function tests, and the characteristics of the pulmonary lesions on chest CT scan, were collected. The maximum standard uptake value (SUVmax) of the target lesions on fluorodeoxyglucose positron emission tomography/CT (FDG PET/CT) was measured before ENB. The final diagnosis for patients not diagnosed by ENB was confirmed by surgical biopsy or follow-up chest CT for at least three months. If primary lung cancer was identified via ENB, the possibility of performing mutational analysis on the samples for EGFR, ALK, and PD-L1 was confirmed by a pathologist (H.S.H.).

### 2.2. Procedure and Clinical Measurements

The ENB system consists of the following three steps [6,14,15]. First, in the planning phase, all subjects should have a chest CT per the protocols recommended by each commercial navigation platform to create a virtual airway map. By multiplanar CT reconstructions on the navigation system, the software automatically creates the closest immediate airway route from the trachea to the marked targets. The next step is the registration phase for synchronizing the sensors between the virtual and actual patient anatomy by pointing out the prominent landmarks such as the carina, second carina, and bronchus during real-time bronchoscopy. Finally, during the navigation phase, the catheter approaches the target lesions along the given route, and whether the approach is appropriate is assessed by the ENB system image in real-time. All procedures were performed with the patients under conscious moderate sedation using midazolam (2–3 mg) and fentanyl (50 mcg) intravenously by two interventional pulmonologists at Asan Medical Center. The SPiN Thoracic Navigation SystemTM (SYS-4230K; Veran Medical, St. Louis, MO, USA) for ENB and a bronchoscope with an outer diameter of 4 mm (P260F) or 6 mm (1T260) (both Olympus Corporation, Tokyo, Japan) were used in this study. For needle aspiration, a 21-gauge (INS-0392; Veran Medical, St. Louis, MO, USA) or a 22-gauge needle (INS-5410; Veran Medical, St. Louis, MO, USA) with a pressure of a20 mL VacLok syringe was used in this study. The size of the forceps (INS-0372; Veran Medical, St. Louis, MO, USA) was 5 mm when the cup was fully open. The ENB in this study was performed by a bronchoscopy expert (C-M.C.) with 15 years of experience and an assistant (W.J.) with 3 years of experience. Furthermore, to ensure the high quality of the procedure, sufficient training and demo cases were performed using the ENB equipment before starting the study. Furthermore, a radiologic expert (MY.K.) with 30 years of experience set the process for optimized images according to the CT protocol provided by the commercial navigation platform (Veran Medical, St. Louis, MO, USA). Additional r-EBUS and ROSE were not used in this study. Procedure-related complications were classified according to their type and severity. Pneumothorax was identified using a post-procedure chest X-ray. A more-than-moderate bleeding event was recorded in cases requiring further management to stop the bleeding, such as applying topical epinephrine or cold saline.

### 2.3. Molecular Testing for Lung Cancer

According to the lung cancer treatment guidelines, the possibility of performing biomarker tests, including those for EGFR, ALK, and PD-L1, on the samples was evaluated by a pathologist (HS. H.) when the specimen obtained by ENB was confirmed as non-small cell lung cancer (NSCLC). For example, in the case of adenocarcinoma, RT-PCR tests to identify EGFR gene mutations (PANAMutyperTM R EGFR kit, Panagene Inc., Daejeon, Republic of Korea), ALK clone D5F3, and PD-L1 clone 22C3 and SP263 immunostaining, were performed. In contrast, in the case of squamous cell carcinoma, only PD-L1 clone 22C3 and SP263 immunostaining was evaluated. The RT-PCR-based EGFR gene mutation test was considered to be eligible if the amount of the remaining specimen within the paraffin block was sufficient for DNA extraction and the tumor purity of the specimen was ≥1% [16]. PD-L1 immunostaining was eligible if ≥100 tumor cells were present within the diagnostic hematoxylin and eosin-stained biopsy tissue slides [17,18]. There are no consensus guidelines for defining the eligibility for ALK D5F3 immunostaining at the present time; as such, the eligibility criteria for PD-L1 immunostaining were adopted for ALK D5F3 immunostaining. If the biomarker tests listed above were already performed, the results were documented.

### 2.4. Statistical Analysis

Categorical variables were expressed as numbers and percentages, and continuous variables were expressed as median (interquartile range (i.e., 25–75% percentile)) or mean ± standard deviation (SD). Definitions of diagnostic yield and accuracy are described elsewhere [19]. To analyze factors affecting diagnostic yield, univariate and multivariate logistic regression models were used, and variables with *p* < 0.1 in the univariate analysis were included in the multivariate analysis. Differences with *p* < 0.05 were considered to be statistically significant. All statistical analysis were performed using SPSS version 21.0 (IBM Corporation, Armonk, NY, USA).

## 3. Results

### 3.1. Baseline Characteristics of the Subjects

Data from 30 patients with 30 nodules were included in the analysis. The median age of the subjects was 63 years and 70% had a history of smoking. Most patients (19/30 (63.3%)) could not undergo PCNB due to vascular structure(s) near the target lung lesions, and seven patients were selected for ENB due to the high risk for pneumothorax from underlying emphysema. Four patients exhibited peripheral lung nodules inaccessible by PCNB (Table 1).

### 3.2. Baseline Characteristics of Lung the Nodules

Approximately one-half of the lung lesions were localized in the upper lobe (17/30 (56.7%)) and the mean diameter was 25.2 ± 7.8 mm. Most nodules were of the solid type and the bronchus sign was found in 20 (66.7%) nodules. Biopsy of 23 nodules was performed using only forceps and four nodules were obtained by needle aspiration in addition to forceps (Table 2).

### 3.3. Results of ENB 

Among the 30 subjects, the approach to the target lesion using ENB in one failed due to a sharp bronchus angle. Thus, ENB enabled the acquisition of target tissue in 29 of 30 cases, corresponding to a success rate of 96.7%, with a diagnostic yield of 58.6% (17/29). After excluding 4 patients with no final diagnosis, among the 25 patients, 17 patients were correctly diagnosed by ENB. Therefore, the diagnostic accuracy was 68% (17/25). Among 17 cases diagnosed via ENB, 14 were primary lung cancer and 3 were benign lesions, including hamartoma, tuberculous granuloma, and inflammatory change(s) (Figure 1). Among the 12 patients not diagnosed by ENB, pathological confirmation via video-assisted thoracoscopic surgery (VATS) or PCNB was achieved in 6, and 2 underwent chest CT follow-up. The final diagnosis in four patients was unknown due to loss to follow-up or death.

### 3.4. Factors Affecting the Diagnostic Yield

Results of the univariate and multivariate analysis are summarized in Table 3. The bronchus sign and total procedure time were identified as significant variables influencing the diagnostic yield of ENB in multivariate analysis. 

### 3.5. Safety 

Procedure-related complications occurred in one patient (Table 4), who experienced pneumothorax that required chest tube insertion, but they recovered quickly and were discharged a few days later. No patients experienced a major bleeding event that required intervention or transfusion. A procedure-related death has not been reported to date.

### 3.6. Molecular Testing Results

Among 14 cases of primary lung cancer confirmed by ENB biopsy, 12 were diagnosed as adenocarcinoma, 1 as squamous cell carcinoma, and 1 as small cell carcinoma. Of the 12 cases of adenocarcinoma, 11 (91.7%) had adequate samples for all molecular tests including EGFR, ALK, and PD-L1. However, one sample exhibiting atypical cells suggested adenocarcinoma was insufficient to obtain a molecular test result (Figure 2). One squamous cell carcinoma was eligible for the PD-L1 assay. Furthermore, we measured the size of the actual tumor tissue to estimate the number of tissue sections that could be produced. Mean small dimensions of the positive tissue sections were 1.46 ± 0.43 mm with 1.5 median number of the tissue sections (range, 1–5).

## 4. Discussion

In this prospective study, we determined the diagnostic yield and safety profile of ENB-guided biopsy. The overall diagnostic yield was 58.6% and the complication rate was 3.4%. We found that the presence of bronchus sign and total procedure time significantly influenced the diagnostic yield. Additionally, specimens obtained by ENB biopsy demonstrated reliable adequacy for molecular testing, including EGFR, ALK, and PD-L1.

The diagnostic yield in this study was comparatively lower than in previous research, which reported yields >60% [10,12]. Several factors may explain this discrepancy. First, it may be particularly relevant to the method of sedation. The detection of target lung lesions can be affected by respiratory motion [20,21]. Because the ENB procedure in most studies was performed on intubated patients under general anesthesia [12,22], patient movement was minimized and their breathing was regular and stable, thus making it easier to perform a biopsy. However, in the present study, we chose moderate sedation using midazolam and fentanyl so the patients were conscious with spontaneous breathing during the ENB procedure. In a meta-analysis of 15 ENB trials, general anesthesia during the procedure demonstrated a better diagnostic yield than conscious sedation (69.2% versus 57.5%, respectively; *p* = 0.02) [19]. Second, although ENB alone was performed in this study, higher diagnostic yields were reported in combination with other techniques, such as ROSE [10,21,23] or r-EBUS [24]. Therefore, we expect that a multimodality approach may help improve the diagnostic yield of ENB and further large prospective studies to confirm the advantage of combining other modalities with ENB also need to be performed. Although it is still difficult to perform ROSE during ENB due to the conditions of Korea, r-EUBS has been used so far. Therefore, it seems possible to conduct a study to identify the benefits of performing ENB with r-EBUS together. In addition, the fact that this was among the first prospective studies of ENB procedures performed in our center could be another reason for the lower diagnostic yield. Accumulation of physician experience with the ENB procedure has been known to be a critical factor affecting the diagnostic yield. Lamprecht et al. noted that the diagnostic yield increased from 80.0% in the first 30 cases to 87.5% after 80 cases [22].

The presence of a bronchus sign has been reported to be one of the most important variables affecting the diagnostic yield [8,12], which is also consistent with the present study. In a study by Webb et al., patients with positive bronchus sign in lesions exhibited a lower degree of displacement between the target lesion and the probe compared to those in whom the bronchus sign was absent (2.6 mm vs. 13.8 mm; *p* < 0.001) [25]. Considering this difference, it would be easier to obtain adequate tissues from a patient with a patent airway leading to the target lesion. Additionally, the current study found an association between diagnostic yield and procedure time. Increased difficulty in approaching lesions resulted in longer procedures, which in turn would decrease the diagnostic yield because it was difficult to obtain adequate sample(s).

When we analyzed the 12 cases that were not diagnosed with ENB, a common feature was that most of the target lesions were not properly approached during biopsy. In particular, only 50% of 12 cases had a bronchus sign. Of six cases without a bronchus sign, only one case used both biopsy tools, forceps and a needle; four cases had a biopsy using forceps; and one case used a needle only. Considering that most of the other ENB studies showing relatively high yields used more than three biopsy tools, if additional biopsy tools such as a needle, brush, and forceps were used together, this could be helpful to increase the diagnostic yield [11].

In patients in the advanced stages of lung cancer, finding target molecules, such as EGFR, ALK, and ROS1, is significant for making treatment decisions; as such, it is necessary to perform molecular testing at the time of initial diagnosis [14]. Therefore, it would be helpful to know whether the small biopsy specimen acquired by ENB-guided biopsy is suitable for molecular testing. In most cases, we were able to obtain primary lung cancer samples of sufficient quality using ENB-guided biopsy to perform molecular testing, including EGFR, ALK, and PD-L1. For confirming the availability of tissue sections for molecular pathology, we measured the number and two-dimensional sizes of the tissue sections which were positive for malignant cells for each case by microscopic examination. The mean diameter of the tissue sections (1.46 ± 0.43 mm) is comparable to the outer dimension of 17-guage needle (1.47 mm) which is routinely used in PCNB. Furthermore, considering the thickness of the ordinary tissue sections (4 μm), a sufficient number of the tissue sections can be produced from the ENB tissue for molecular analysis. Therefore, in a similar manner to the lung percutaneous needle biopsy specimens, ENB tissue sections can be potentially used for essential molecular tests. Of course, because a biopsy specimen is only a limited portion of the total tumor tissue, the test may be affected by tumor heterogeneity, such as PD-L1 analysis, and, therefore, can be less reliable. However, considering previous studies that have reported a high concordance rate of the PD-L1 assay between surgical specimens and paired small biopsy samples by EBUS-TBNA, bronchoscopic biopsy, or PCNB, samples obtained by ENB could also be used effectively [26,27]. As our research has shown, molecular testing, including clonal-like EGFR mutation and ALK translocation analysis, has been reported to be successful for most NSCLC tissues obtained by ENB [11,28]. Especially, in the present study, most (11/13 (91.6%)) NSCLC patients diagnosed using ENB biopsy were in the early stage(s) of lung cancer (i.e., stage I–II), even with no lymph node metastasis. It is clinically meaningful to emphasize that sufficient samples for histological subtyping and molecular testing can be obtained by ENB in early-stage lung cancer as well as in advanced stage(s) with larger tumor burden. Despite the early stage of NSCLC, patients who are inoperable due to older age or low baseline lung function with underlying lung disease are usually treated with stereotactic body radiation therapy [5,29]. However, local recurrence and regional nodal recurrence could occur after planned treatment [30]. Therefore, if it is possible to perform molecular testing using a less invasive method, such as ENB, we could choose more effective therapeutic options in those with a recurrent metastatic stage as well.

ENB appears to be much safer than PCNB, with the risk for complications reported to be approximately 3%, with the most common complication being pneumothorax [31]. In the present study, we also demonstrated an acceptable safety profile, with only one procedure-related complication (i.e., pneumothorax). Considering that suspicious lung nodules are frequently found in heavy smokers with underlying lung disease—thus increasing the risk for complication(s)—the use of ENB should be encouraged as an alternative to other percutaneous methods.

The present study had a few limitations, the first of which was its single-center design and relatively small case series. Nevertheless, it is valuable in that it was the first prospective study using the SPiN Thoracic Navigation System, which enables real-time respiratory gating, to verify the utility of ENB in Korea, where diagnostic bronchoscopy procedures are highly developed. Second, it was difficult to know exactly the difference in diagnostic yield and accuracy between non-ENB and ENB methods because this study is a single-arm non-comparative study. Although PCNB has shown a high diagnostic yield and accuracy up to 90% and 96%, respectively, widespread use of this approach is limited due to the relatively high risk of complications [1,32,33]. Radial EBUS is also available for biopsy of peripheral lung nodules with the advantages of a 70% high diagnostic yield, accuracy of 72%, and approximately 2–3% complication rate [1,34,35]. However, according to a meta-analysis, when the lesion size decreases below 3 cm, the diagnostic yield tends to decrease rapidly into the 50–60% range [34]. The diagnostic yield and accuracy of r-EBUS is comparable with the results of this study when considering that the size of most lesions was less than 30 mm. In addition, this prospective study is particularly meaningful in that a nearly 60% diagnostic yield was obtained despite only targeting peripheral lung lesions that were difficult to biopsy by other methods. Third, because most of the patients diagnosed with lung cancer by ENB in this study were in an early stage of the disease, results of molecular testing in patients with advanced-stage disease were lacking. In particular, among patients who fail initial EGFR-TKI treatment and require additional re-biopsy, if the feasibility of ENB-guided biopsy for identifying T790M mutations could be evaluated, a wider range of patients would benefit. Therefore, further investigations are required to verify the utility of ENB for patients in the advanced stage(s) of lung cancer.

## 5. Conclusions

This prospective study demonstrated an acceptable diagnostic yield and safety profile for ENB performed in 30 patients at a single institution in the Republic of Korea. The diagnostic yield was associated with the presence of a bronchus sign and the procedure time. Furthermore, our results demonstrated that the samples obtained using ENB were sufficient to perform molecular analysis. We anticipate further larger-scale, prospective, multicenter studies to analyze more cases in various environments.

## Figures and Tables

**Figure 1 diagnostics-11-01432-f001:**
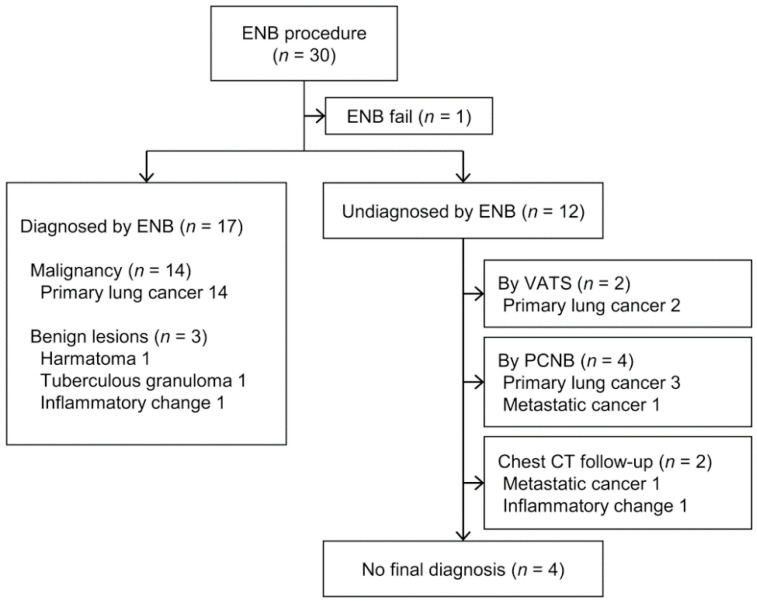
Flow diagram for 30 electromagnetic navigation bronchoscopy (ENB) cases and the final pathological results.

**Figure 2 diagnostics-11-01432-f002:**
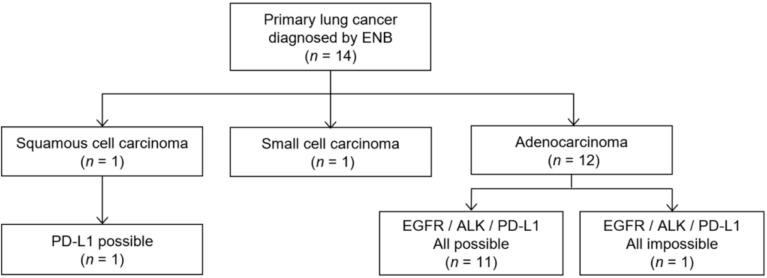
Flow diagram of molecular testing results in lung cancer diagnosed using electromagnetic navigation bronchoscopy (ENB).

**Table 1 diagnostics-11-01432-t001:** Baseline characteristics of the study participants.

	Number (%)
Total participants	30
Age in years, median (range)	63 (56–72)
Male sex	21 (70)
Smoking	
Never	9 (30.0)
Ex-smoker	13 (43.3)
Current smoker	8 (26.7)
Pulmonary function	
FVC, % pred, mean ± SD	85.8 ± 12.1
FEV1, % pred, mean ± SD	81.2 ± 114.9
DLco, % pred, mean ± SD	78.6 ± 22.1
Cause of ENB	
Emphysema	7 (23.3)
Vascular structure	19 (63.3)
Inaccessible	4 (13.3)
Intravenous sedation	30 (100)
Total procedure time, mean ± SD, min	22.7 ± 11.8

Data are presented as number (%), mean ± standard deviation (SD), or median (interquartile range). FVC, forced vital capacity; FEV1, forced expiratory volume in one second; DLco, diffusing capacity of the lung for carbon monoxide; IV, intravenous.

**Table 2 diagnostics-11-01432-t002:** Baseline characteristics of the pulmonary nodules.

	Number (%)
Total nodules	30
Location	
Right upper lobe	8 (26.7)
Right middle lobe	4 (13.3)
Right lower lobe	5 (16.7)
Left upper lobe	9 (30.0)
Left lower lobe	4 (13.3)
Size, mm ± SD	25.2 ± 7.8
Type	
Solid	22 (73.3)
Ground-glass opacity	1 (3.3)
Partially solid	5 (116.7)
Consolidation	2 (6.7)
Bronchus sign present	20 (66.7)
Metabolic activity in PET, mean SUVmax ± SD (*n* = 27)	7.7 ± 6.6
Distance from visceral pleura, mm ± SD	9.7 ± 8.6
Biopsy tool used	
Forceps	23 (76.7)
Needle	3 (10.0)
Forceps + needle	4 (13.3)

SD, standard deviation.

**Table 3 diagnostics-11-01432-t003:** Univariate and multivariable analysis of factors associated with the diagnostic yield.

	Univariate Analysis		Multivariate Analysis	
	OR (95% CI)	*p*-Value	OR (95% CI)	*p*-Value
Lower lobe lesion	0.909 (0.184–4.500)	0.907		
Nodule size, mm	1.079 (0.967–1.205)	0.175		
Solid type nodule	0.367 (0.060–2.252)	0.279		
Peripheral location	1.556 (0.256–9.469)	0.632		
Bronchus sign	4.667 (0.866–25.136)	0.073	15.874 (1.568–160.6)	0.019
Metabolic activity, SUVmax	1.032 (0.911–1.168)	0.620		
Distance from Visceral pleura, mm	0.979 (0.899–1.067)	0.629		
Total procedure time, min	0.893 (0.799–0.999)	0.048	0.842 (0.718–0.987)	0.034
Biopsy device				
Forceps only	Ref			
Needle only	0.286 (0.022–3.669)	0.336		
Forceps + needle	0.571 (0.067–4.875)	0.609		

CI, confidence interval; OR, odds ratio. Variables with *p*-value < 0.1 were included in multivariate analysis.

**Table 4 diagnostics-11-01432-t004:** Safety profile of ENB.

Adverse Event	Number (%)
Overall	1 (3.4)
Pneumothorax	1 (3.4)
Need for chest tube insertion	1 (3.4)
Bleeding	0 (0)
Respiratory failure	0 (0)
Death	0 (0)

## Data Availability

The datasets used and/or analyzed during the current study are available from the corresponding author on reasonable request. Informed consent was obtained from all subjects.

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
