# Peer review of "Diagnostic Performance of Electromagnetic Navigation Bronchoscopy-Guided Biopsy for Lung Nodules in the Era of Molecular Testing"

_diagnostics, 2021, doi:10.3390/diagnostics11081432_

Round 1

Reviewer 1 Report

In our opinion, you should mention the indication and contraindication for ENB, taking into consideration the patient selection (inclusion criteria).

Second, you should detail the ENB technique in the method section. 

In the 151 row, we do not understand the accuracy of 68% (17/25), how do you get this number? Did you exclude from the total group, the 4 patients with no final diagnosis? please explain more detailed. 

In the 229 row, we do not understand the numbers.. 11/13 patients were in the early stages of lung cancer.. how did you get this number? Did you exclude the patient without EGFR/ALK/PDL-1? Please explain more detailed. 

Reviewer 2 Report

This is an important contribution to diagnostic procedures but doesn't emphasize the importance of high quality training and need for radiological expertise and equipment. 

Reviewer 3 Report

The authors conducted a prospective study of 30 patients who underwent transbronchial lung biopsy supported by electromagnetic navigation bronchoscopy (ENB) to determine the accuracy and safety of the examination, and whether molecular pathology, which has become particularly important in recent years, could be retrieved. They reported that TBLB supported by ENB had a diagnostic accuracy of 68%, and in 92.3% of cases where malignant tumors could be collected, samples could be retrieved for molecular pathology.

I thought this study contained some severe problems, so I would like to point out as below.

  1. I could not understand the purpose of conducting this study as a perspective study. This study is not a comparison between ENB guided biopsy and non-ENB guided biopsy. Also, if the endpoint is the diagnostic accuracy rate and the incidence of adverse events in ENB guided biopsy, there is not much need for small prospective studies, and the information presented in large retrospective cohort studies and meta-analyses is sufficient. I think that one of the important recommendations of this prospective study is the consideration of cases that were failed to diagnosis. However, little of this consideration is given in this paper. I propose that the significance and necessity of conducting this study as a prospective study be well described in the introduction, followed by an extensive discussion of the new scientific findings of this study.

  1. If you want to emphasize the usefulness of ENB guided biopsy, I think you need to show that the diagnostic accuracy of ENB guided biopsy is higher than that of non-ENB guided biopsy. Although this study is not a comparative study, we believe it is important to consider the diagnostic accuracy of ENB guided biopsy in general terms. The average tumor diameter in the population of this study was less than 30 mm, and it is expected that the diagnostic accuracy of non-ENB guided biopsy is low. I think it is necessary to discuss the diagnostic accuracy of this study in comparison with the previously reported non-ENB guided biopsy.

  1. In this study, one of the endpoints was whether molecular pathological examination was possible. The size and number of sections collected have a great impact on whether a tissue section is available for molecular pathology or not. Also, for the tissue sections obtained, how many blocks were made when they were paraffin embedded may also have an influence. Due to the lack of such information in this study, it is not possible to determine whether ENB guided biopsy is useful for the collection of tissue samples that can be extracted for molecular pathology.

  1. I think the size of the forceps and aspiration needle used will also affect the size of tissue sections which can useful for molecular pathology. The thickness of the forceps or aspiration needle used for biopsy in this study should also be added to the Materials and Methods section.

  1. Whether r-EBUS or ROSE was used in addition to ENB during biopsy has a significant impact on the diagnostic accuracy rate. The discussion mentions that r-EBUS and ROSE were not used in this study, but please make sure that this is also indicated in the Materials and Methods section.

  1. Isn't the number of cases where forceps biopsies were performed listed in line 142 a mistake of 27?

  1. Please correct the following words, if necessary.

              harbouring           harboring            

              tumour                tumor

              haematoxylin       hematoxylin

              analyse                analyze 

              anaesthesia          anesthesia

Round 2

Reviewer 3 Report

The authors responded well to my rather severe comments last time.

Although my lack of understanding may have been one of the reasons, I believe that the authors clarified the significance of this study, which was slightly difficult to understand, and revised the manuscript to provide a better understanding.

In addition, I felt that it was significant to report the accuracy of the examination while the sedation method was different from the Western method.

(This is not emphasized much in the manuscript, but I thought it might be worth highlighting as a strength point of the study, if the authors considered.)

I would like to mention one additional opinion,

In my previous point 5, I commented on the combined use of ENB and r-EBUC or ROSE. The authors discuss the possibility of improving the diagnostic accuracy by using these multiple modalities. The authors also mentioned that the diagnostic accuracy of ENB in this study was slightly lower than that of previous studies. It may be useful to consider what attempts should be made in the future to improve the diagnostic accuracy in specialized facilities in Korea, such as the authors' facility, and whether further prospective studies on this are required or not. I am not sure if this statement is necessary to emphasize the significance of this study, so please consider additional comments if the authors believe that it is necessary.
